# Prospective study of change in liver function and fat in patients with colorectal liver metastases undergoing preoperative chemotherapy: protocol for the CLiFF Study

Kat L Parmar,[1,2] Derek O'Reilly,[3] Juan W Valle,[1,4] Michael Braun,[4] Jo H Naish,[5] Steve R Williams,[6] William K Lloyd,[6] Lee Malcomson,[1,7] Katharine Cresswell,[8] Colin Bamford,[9] Andrew G Renehan[1,7]

For numbered affiliations see end of article.

**Correspondence to**
Kat L Parmar;
kat.parmar@manchester.ac.uk

## ABSTRACT

**Introduction** Preoperative chemotherapy in patients undergoing resection for colorectal liver metastases (CLM) improves oncological outcomes. However, chemotherapy-associated liver injury (occurring in two patterns: vascular and fat deposition) is a real clinical concern prior to hepatic resection. After major liver resection, regeneration of the residual liver is a prerequisite for recovery and avoidance of liver failure, but this regenerative capacity may be hindered by chemotherapy. Thus, there is a need to predict for this serious complication. Over the past two decades, several tests and derived indices have been developed, which have failed to achieve clinical utility, mainly as they were indirect measurements of liver function. Here, we will use a novel test of liver function (the liver maximum capacity (LiMAx) test), and measure liver fat using MRI.

**Methods and analysis** This prospective study will assess changes in liver function longitudinally, measured by the LiMAx test, and liver fat, measured by advanced MRI using both MR spectroscopy and the modified Dixon method, in up to 35 patients undergoing preoperative chemotherapy for CLM. The primary outcomes will be the changes in liver function and fat compared with baseline prechemotherapy measurements. Secondary outcome measures include: routinely measured liver function blood tests, anthropometric measurements, postoperative histology and digital quantification of fat, postoperative complications and mortality and quality of life.

**Ethics and dissemination** The study was approved by a National Health Service Research Ethics Committee and registered with the Health Research Authority. Dissemination will be via international and national conferences and the National Institute for Health Research network. Manuscripts will be published.

**Trial registration number** This study is registered online at www.clinicaltrials.gov (registration number NCT03562234).

### Strengths and limitations of this study

► This is an evaluation of the effects of preoperative chemotherapy on liver function and fat using novel techniques, addressing the important clinical issue of liver injury prior to hepatic resection.

► These non-invasive techniques permit repeated measurements to assess change in liver function and fat over time, permitting interpretation of causality—this has not been possible in the past.

► Chemotherapy-related changes in liver fat and function will be assessed for reversibility and correlated with both standard assessment measures and postoperative histology, providing insight into the scale of the injuries prior to the postchemotherapy recovery period.

► Changes and recovery in liver fat and function will be correlated with postoperative morbidity and mortality.

► A potential limitation is that this is a single-site observational study with a relatively small sample size.

Death is usually due to metastatic disease, with the most common site being the liver (colorectal liver metastases—CLM). There is an incidence of 15%–20% CLM at initial diagnosis of CRC[2] with up to 50% of patients ultimately developing CLM.[3–6]

Surgical resection of CLM is the only single-modality therapy associated with cure and remains the mainstay of treatment.[7] The resectability of CLM depends on the volume and function of the future liver remnant (FLR). Volume alone is not an ideal assessment of liver functionality as it cannot reliably predict outcome.[7–12] FLR function is decreased in patients with parenchymal disease, despite being of equal volume to patients with healthy livers,[13–15] and liver disease may result in impaired liver

## BACKGROUND

### Colorectal cancer (CRC) is the second most common cause of cancer death in the UK.[1]

regeneration.[7 11 16–18] In the presence of impaired liver function, the surgical strategy may need to be adapted, either to increase the volume of the FLR or perform a less extensive resection.[2 19] Accurate preoperative assessment of liver function is, therefore, crucial.[20]

With best supportive care, the median survival of patients with CLMs is 6–13 months (where 5-year survival is rare).[21] In patients fit enough to receive combined hepatic resection and chemotherapy, the 5-year survival improves to 50%.[22–27] Preoperative chemotherapy is used in the setting of both resectable and unresectable CLM, with the intentions of (1) improving survival (neoadjuvant approach) in operable CLM and (2) converting inoperable CLM into operable (downsizing approach). Approximately 70% of patients presenting with CLM are initially deemed unresectable,[28] and after systemic chemotherapy, up to 40% of patients may be judged resectable.[29–34] Outcomes in these 'downsized' cases can be similar to those initially presenting with resectable disease.[35 36] In the setting of operable CLM, the randomised EPOC Trial reported a 7.3% improvement in 3-year progression-free survival with addition of chemotherapy versus resection alone for CLM, and highlighted the issue that organ-specific resectional surgery does not address the systemic process, leading to high postoperative recurrence rates.[37]

The main concern regarding the preoperative administration of chemotherapy for CLM is direct pathologic changes to the liver, known as chemotherapy-associated liver injury (CALI). CALIs comprise two main types of liver injury: vascular changes and fatty changes. These injuries are not mutually exclusive; patients who have developed fatty changes can simultaneously develop vascular changes.[7] There are many inconsistent reports on the types of liver injury seen and their associations with specific chemotherapy regimens.[11 30 37–58] There is considerable heterogeneity in the methods used to assess changes to the liver parenchyma. Earlier studies used CT scans and liver biopsies[38–41] and later studies used postoperative histology, reported using a variety of scoring systems.[11 30 37 42–58] Radiation exposure resulting from CT scans and the invasive nature of liver biopsies mean that neither technique is suited to providing repeated measurements, and postoperative specimens provide only a one-off assessment of the liver parenchyma. None of these assessment tools are, therefore, ideally suited to measuring change over time.

The link between CALIs and postoperative morbidity and mortality remains unclear with reports to date being mixed.[30 37 42–45 47–53 55–59] Some studies such as the one by Karoui et al[47] have demonstrated that CALIs increased the risk of postoperative liver failure (11% vs 0%), with others such as Vauthey et al[48] demonstrating increased postoperative mortality (increased 90-day mortality in patients with steatohepatitis 14.7% vs 1.6%, p=0.001; steatohepatitis associated with preoperative irinotecan 20.2% vs 4.4%, p<0.001).

Postoperative morbidity and mortality are often related to inadequate function of the remnant liver, resulting in postoperative liver failure—an important cause of mortality after partial liver resection.[14 60 61] A meta-analysis estimated the overall incidence of liver failure after hepatectomy to lie between 0.7% and 9.1%,[34] with most recent studies reporting liver failure rates within this range.[58 62] The improved detection of CALIs, assessment of their impact on postoperative outcomes and preoperative assessment of the FLR are, thus, important clinical issues. Several tests and derived indices are in use, including conventional blood parameters of liver function [liver enzymes, albumin, bilirubin and International Normalised Ratio (INR)], Child-Pugh Classification, scintigraphy, galactose elimination test and indocyanine green (ICG) clearance rate. These assessments are suboptimal as they comprise indirect measurements of liver function.

Advances in chemotherapy and hepatic surgery have expanded the pool of candidates for potentially curative hepatic resection for CLM,[7] many of whom will undergo preoperative chemotherapy. Recognition of CALIs emphasises the need to fully assess each patient and use individualised planning, giving attention to the planned extent of resection, the choice of chemotherapy regimen and the potential consequences on the hepatic parenchyma. Future challenges include the refinement of liver function assessment and the establishment of better methods for evaluation and diagnosis of postchemotherapy liver injury.[7]

This study aims to address these challenges by uing the following two novel techniques for assessing liver function and liver fat, measuring these repeatedly before and after chemotherapy, and correlating these changes to postoperative outcomes. Performing repeated measurements over time of both liver fat deposition and liver function during and after the cessation of chemotherapy will permit better interpretation of causality and potential reversibility. This will address the mechanistic question shown in figure 1: is chemotherapy-induced hepatic fat accumulation driven through liver injury as shown in Hypothesis A, or unrelated as in Hypothesis B?

Understanding these relationships and making use of the observation that preoperative chemotherapy is a model of an extreme accelerated phenotype of liver fat deposition may also help to address some of the underlying mechanistic questions in obesity-related cancers.

## The liver maximum capacity (LiMAx) test

The LiMAx test was developed as a novel clinical evaluation of liver function, aimed at overcoming the difficulties in accurately assessing preoperative liver function prior to hepatic resection.[63] It is based on the metabolism of a $^{13}$C-labelled substrate by the hepatocyte-specific enzyme system P450 CYP1A2, the activity of which is not influenced by drugs or genetic variations,[64] is distributed ubiquitously throughout the liver,[65] and shows a clear discrimination between normal and abnormal liver function independent of cholestasis.[66] The intravenous administration of $^{13}$C-methacetin results in a significant

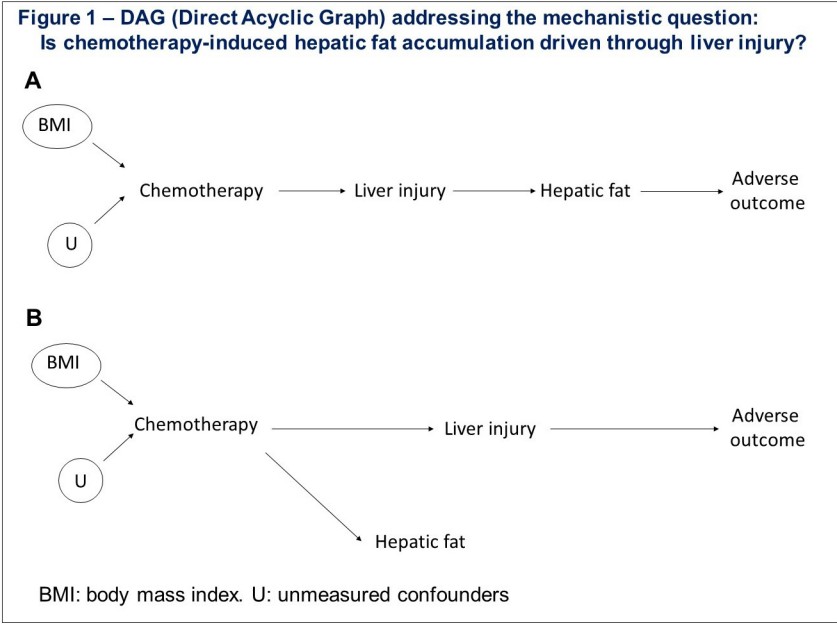

**Figure 1** Direct acyclic graph.

alteration of the normal expired breath $^{13}CO_2$:$^{12}CO_2$ ratio[67] detected by breath analysis performed by an infrared spectroscopy-based FLIP device.

The LiMAx test was initially used in preoperative and postoperative hepatectomy assessment, where resection of a specific percentage of functional liver volume was found to lead to an equivalent reduction in LiMAx value.[68] The LiMAx test was found to represent an accurate surrogate of liver function capacity,[68] to be an accurate predictor of postoperative liver failure and mortality[63 68–70] and to be unaffected by age and gender.[71]

The LiMAx test has been evaluated as a tool with which to diagnose CALIs. LiMAx was found to be superior to other methods of assessing liver function both in detecting CALIs and their resolution, when compared with standard biochemical liver parameters (INR, bilirubin and aspartate transaminase) and ICG clearance rates.[72 73] However, it has never been used repeatedly in the setting of CLM or correlated with postoperative outcomes.

### Advanced MRI

Traditional assessment methods failed to provide an ideal measurement tool for liver fat content (hepatic fat fraction—HFF). Ultrasonography scan (USS) and CT have the ability to demonstrate fatty infiltration; however, they are unable to quantify HFF.[74–77] USS is preferred for qualitative assessment because CT sensitivity is time dependent and protocol specific[78] and is not suitable for use in longitudinal monitoring due to its reliance on ionising radiation.[79] On the other hand, USS is low cost, safe, and readily accessible.[79] However, USS has limited repeatability and reproducibility, and the outputs have only modest correlation with biopsy results.[78 80] Liver biopsy provides a quantitative method for the assessment of fatty liver disease; however, this is an invasive procedure

hampered by sampling errors as fatty infiltrations can be unevenly distributed throughout the liver,[81–83] significant interobserver variability[84] and potential complications, such as bleeding and mortality.[85]

MRI using standard pulse sequences is insensitive to diffuse fatty infiltrations.[86–88] Techniques have been developed with the aim of further exploiting the differences in resonance frequencies between water and fat proton signals in order to assess HFF with increased accuracy, including MR spectroscopy (MRS)[89] and the modified Dixon technique (mDixon).[90]

MRS uses a much larger sample of the liver as compared with a typical biopsy sample, minimising the chance of sampling error, and is sensitive enough to detect small amounts of fat that may be histologically undetectable.[91] It is considered to be the optimal non-invasive method for the accurate assessment of liver fat,[92] being the technique shown to produce the most reliable and reproducible measurements.[79 81 91 93–98] MRS has been validated as an accurate technique against the histological evaluation of fatty liver changes in multiple studies.[81 93–96 99 100] However, MRS has not achieved widespread clinical use despite the undisputed strengths of this technique. This may be due to poor availability of expertise and equipment outside the research setting.[90]

Techniques of quantitative MRI have recently been introduced for the acquisition of proton density fat fraction maps covering large portions of the liver. These mDixon show promising results in assessing HFF in comparisons with histological scoring systems[101–103] and correlate well with MRS.[90]

The Manchester Obesity and Cancer Research Group run a long-term research programme developing non-invasive imaging techniques to quantify liver fat. We have

previously validated the use of MRS and the mDixon method against digital histological determination of fatty infiltration in postsurgical specimens, quantifying and characterising fat deposition in patients with CLM undergoing resection.[104 105] These previous studies were cross-sectional, whereas the current study will evaluate changes over time in order to better assess their relationship to chemotherapy and their potential reversibility.

The proposed MRI protocols in the current study will be as follows:

### MR spectroscopy
Single-volume 1H spectra are acquired using Stimulated Echo Acquisition Mode spectroscopy [(Repetition Time (TR)=2000 ms, Echo Time (TE)=10 ms, Flip Angle (FA)=90°]. Sixteen dynamic scans are acquired without averaging for a $15\,mm^3$ voxel of interest (VOI) with a 32 s duration. This is repeated three times with VOIs positioned to avoid CLM and partial volume averaging with large blood vessels and bile ducts.

### Modified Dixon techniques
A six-echo three-dimensional gradient echo sequence is used, with first TE at 0.92 ms and a delta of 0.7 ms (TR=5.3 ms, FA=5°). Full liver coverage is acquired with 67 axial slices and $3\,mm^2$ in-plane and $6\,mm$ through plane resolution, interpolated to $2\,mm^2$ in-plane and $3\,mm$ through plane. Parallel acquisition used with an SENSE (Sensitity Encoding) factor of 1.5 in the Anteroposterior (AP) plane. The full volume is acquired in a single breath hold for a duration of 12 s. Quantitative fat fraction maps are calculated using a dedicated scanner software package. A representative fat fraction for each subject is calculated using the average from three liver-tissue regions of interest positioned on a central slice, using the same criteria as for the spectroscopy.

### Aims
To prospectively assess the changes in liver function and fat resulting from the administration of preoperative chemotherapy prior to hepatic resection for CLM (Change in Liver Function and Fat—the CLiFF Study). These changes will be assessed repeatedly using the novel techniques of the LiMAx test and advanced MRI as demonstrated in figure 2, then correlated with standard assessments and postoperative outcomes. To our knowledge, there is no prospective evaluation of this type to date.

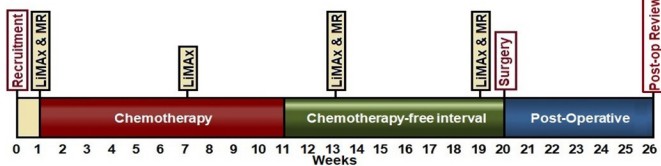

**Figure 2** Study participant timeline. LiMAx, liver maximum capacity.

### Primary outcome measures
1. Change in liver function measured by the LiMAx test performed using the FLIP device, comparing baseline measurements with measurements during and after chemotherapy.
2. Change in liver fat as a percentage (HFF) measured by advanced MRI (using both MRS and the modified Dixon method), comparing baseline measurements with measurements during and after chemotherapy.

### Secondary outcome measures
1. Changes in routinely collected liver enzymes by blood testing.
2. Digital histological quantification of intrahepatic fat.
3. Correlations with BMI and image-determined anthropometric measurements, for example, subcutaneous adipose tissue, visceral adipose tissue volumes and muscle mass.
4. Grade of steatosis scores [the Non-alcoholic Fatty Liver Disease Activity Score proposed by the Non-alcoholic steatohepatitis Clinical Research Network (NAS–CRN)] from routine histology reporting.
5. Postoperative outcomes, including perioperative mortality and morbidity (Clavien-Dindo classification, Comprehensive Complication Index and International Study Group for Liver Surgery complications).
6. Change in health-related quality of life by measurement of EQ-5D Score.

## METHODS
### Study design
This is a non-randomised prospective observational study of up to 35 patients with CLM.

### Study setting
The study is sponsored by the University of Manchester and will be performed at The Christie National Health Service (NHS) Foundation Trust and the University of Manchester Wolfson Molecular Imaging Centre (WMIC). Participant identification will take place at the joint Christie–Manchester Foundation Trust hepatopancreatobiliary (HPB) multidisciplinary team (MDT) meeting. Team members will screen all patients for whom an MDT decision is made that a course of preoperative chemotherapy is indicated prior to potentially curative hepatic resection of CLM.

### Institutions
From October 6 2014, a single Improving Outcomes Guidance compliant HPB Service has been located at the Manchester Royal Infirmary site at Manchester University NHS Foundation Trust. It hosts the regional specialist HPB MDT meeting. All patients selected to undergo neoadjuvant chemotherapy prior to hepatic resection from this MDT will receive this treatment via the oncology service delivered from the Christie Hospital.

The Wolfson Molecular Imaging Centre is part of the University of Manchester. This centre provides

state-of-the-art imaging facilities and radiological expertise in the advanced imaging techniques of MRS and mDixon. The Philips 1.5T scanner at WMIC was upgraded to provide mDixon in addition to the established MRS.

## Investigators

The CLiFF Study is led by a dedicated clinical research fellow, as part of a University-registered thesis. The supervisory team provide the relevant expertise in surgical cancer research, hepatobiliary surgery, medical oncology and imaging science, with additional research and statistical support available as required.

## Inclusion criteria

Patients with liver metastases of colorectal origin undergoing preoperative standard chemotherapy, with the following criteria:

1. Histologically verified adenocarcinoma of the colon or rectum with radiological evidence of potentially resectable liver metastases.
2. No evidence of unresectable non-hepatic metastatic disease.
3. Adequate haematological and hepatic function defined by: haemoglobin $\geq100\,g/L$, white blood count$\geq3.0{\times}10^9/L$, absolute neutrophil count (ANC) $\geq1.5\,x\,10^9/L$, platelet count $\geq100\,000/mm^3$, total bilirubin $<30\,\mu mol/L$, serum aspartate aminotransferase (AST), alanine transaminase (ALT) and alkaline phosphatase $\leq5$ times upper limit of normal.
4. Written informed consent and able to attend long-term follow-up.

## Exclusion criteria

1. Presence of a medical or psychiatric condition that impairs their ability to give informed consent.
2. Presence of any other serious uncontrolled medical conditions.
3. Evidence of unresectable non-hepatic metastatic disease.

## Patient identification

Patients will be screened for inclusion initially via discussion at the specialist HPB MDT meeting and by referral to the Medical Oncology service. Subsequent to an MDT decision that preoperative chemotherapy is indicated prior to potentially curative resection of CLM, the CLiFF Study team will screen for the remaining inclusion criteria. Potential participants will be invited to join the study via the provision of verbal and written information regarding the CLiFF Study. Confirmation of the wish to participate will subsequently be confirmed and informed consent taken prior to beginning chemotherapy.

## Data collection

Participants will undergo assessment of liver function by LiMAx testing and HFF by advanced MRI at the times outlined in the study schedule shown in online supplemental appendices 1 and 2. All clinicians involved in the medical and surgical treatment of the study participants will be blinded to the results of the LiMAx testing and MRI. The study MR images will not be reported by a clinical radiologist. All other aspects of treatment will remain unchanged and proceed as per standard care.

Data collection will include LiMAx tests and MR results, in addition to the results from all other assessments outlined in online supplemental appendix 2. The LiMAx tests and MR scans will be performed directly by the clinical research fellow. The remaining information will be taken from the medical notes, comprising the results from routine oncology outpatient assessments, the postoperative histology report and the postoperative outcomes following hepatic resection. Informed consent will be taken for the research team to access this information from the medical records.

## Data management

The NHS code of confidentiality will be observed, with only the clinical care team and team directly involved in the research having access to any identifiable data. All data will be pseudonymised with a unique identifier and stored in a secure encrypted database. A key to patient identifiers will be stored in a separate encrypted document. Data stored by the research team will, therefore, not contain patient identifiers. Data will be collected and stored in accordance with the General Data Protection Regulation and Data Protection Act 2018. The University of Manchester, as data controller for this study, takes responsibility for the protection of any personal information collected by this study. All researchers will be appropriately trained in data protection.

## Statistical analysis and power calculation

Our previous work reported a mean (±SD) HFF of 4 (±2)%.[105] Following chemotherapy, we estimate at least a 50% increase in HFF that is, increase to mean (SD)=6%±2.5%. We have planned a recruitment of at least 35 patients. As attrition (for multiple reasons) is common in this type of study, of the 35 patients, we conservatively estimate that 25 patients will have complete data from at least 2 MR evaluations. The resultant power is 98% (alpha: 0.05; within-person correlation: 0.7).

For the LiMAx testing, which is more frequent in the study schedule, we estimate at least 32 patients (10% attrition rate) will have complete data from two LiMAx tests. Based on the Lock et al study in patients undergoing preresection chemotherapy[73] (mean (±SD): 340±95 vs 391±82 µg/kg/hour), the resultant power is 84% (alpha: 0.05; within-person correlation: 0.7).

Statistical support will be provided where required by the University of Manchester Cancer Data Science Team. The main analysis will focus on intrapatient change in both liver function (measured by the LiMAx test) and HFF (measured by advanced MRI), which occurs subsequent to chemotherapy, and their recovery prior to surgery. These changes will be related to subsequent clinical course, CT volumetry and histopathological analysis of the resected liver. We will explore the use of

analysis of variance (ANOVA) methods and mixed-effects methods to account for within-person correlations. For each timepoint, standard approaches to categorical ($\chi^2$ test) and continuous (Mann-Whitney U test) variables will be used. Correlation matrices will be constructed to assess relationships between LiMAx and HFF with other continuous variables using Spearman's Rank Correlation Coefficient.

To account for multiple testing, a p value of <0.01 will be considered to indicate statistical significance. Predictors of HFF at each timepoint will be explored using multivariable linear regression models. To reduce the anticipated right skewness of the distributions, logarithmic transformation will be explored.

Because of the large number of variables and the anticipated high levels of correlations, we will use factor-cluster methods. There will be at least five clusters: patient-related factors; routine blood measures; CT-derived anthropometrics; LiMAx readouts and MR readouts. Separate models will be developed for each cluster and significant (p<0.05) variables selected for the final model.

### Anticipated recruitment

At the specialist HPB MDT, there are approximately 120 resections per year performed for CLM. Approximately half of these patients will receive preoperative chemotherapy. Of these 60 patients, 40 will have their chemotherapy delivered at the oncology centre and will be potentially eligible for recruitment. Over the past 4 years, the numbers of liver resections performed for CLM has consistently increased year-on-year. We, therefore, anticipate that the recruitment target of 35 patients to be recruited and followed up within 2 years is achievable.

### Quality assurance

The quality of this study has been assessed by the following means:

► Departmental review within the University of Manchester.
► Peer review by professionals with relevant expertise (clinical trialists, statisticians, surgeons, oncologists and imaging scientists).
► Review by the Research Practice Governance Team at the University of Manchester (Sponsor Institution) and the Research & Development team at the Christie NHS Foundation Trust (host).
► Peer review arranged by the North West Innovation Service as part of a successful competitive funding application.
► Review by the Royal College of Surgeons research department as part of a successful competitive application for a research fellowship.
► Independent peer review as part of the registration process for ClinicalTrials.gov, a publicly accessible database of worldwide research studies, maintained by the National Library of Medicine at the National Institutes for Health (USA).

**Table 1** Study timetable

| Time period | Date |
| --- | --- |
| Study set-up and approvals | 01 April 2018 to 01 September 2018 |
| Participant recruitment | 01 October 2018 to 01 April 2020 |
| Data collection | 01 April 2020 to 01 October 2020 |
| Data analysis and write-up | 01 October 2020 to 31 March 2021 |

### Study timetable

The anticipated study timetable is outlined in table 1.

## ETHICS AND DISSEMINATION
### Ethical approval

Ethical approval for this study was granted by the National Health Service North West Research Ethics Committee following the meeting held on 27 July 2018 (REC Reference 18/NW/0531).

### Registration

This study was registered, reviewed and approved by the Health Research Authority. It has been registered with the sponsor institution (University of Manchester) and approved by their Research Practice Governance Team. The host institution (Christie NHS Foundation Trust) Research & Development team have issued approval of their capability and capacity to host this study.

### Patient and public involvement

The Public Programmes Team at the Manchester University NHS Foundation Trust hosted a discussion group for the CLiFF Study with six members of the National Institute for Health Research Manchester Biomedical Research Centre funded Cancer Research Advisory Panel on 24 May 2018. The six attending members were all patients/carers with personal experience of CLM treatment. This event was attended by the principle investigator to actively seek and hear the views of the patient and public representatives. The group unanimously found that this area of research was important, found no feasibility or acceptability issues with the study and thought it likely that they would take part if asked. The following modifications were made in response to the session:

#### Amendments to participant information sheet

All group suggestions were incorporated, including the requested addition of a clearly highlighted opening statement explaining that treatment would not be modified as the result of participating in this study.

#### Training to use long-term intravenous access devices

Patients who had previously undergone preoperative chemotherapy for CLM expressed frustration that while many of them had long-term intravenous access devices

for the duration of their treatment, inadequate training to access these devices resulted in additional exposure to needles. They requested that where possible, these long-term devices be used for the intravenous injection component of the LiMAx test. In response to this request, the principal investigator underwent additional training to ensure that long-term intravenous access devices could be used for the LiMAx tests performed in this study.

One patient member of the Cancer Research Advisory Panel contributed substantially and subsequently became a coauthor of this protocol paper (CB).

## Dissemination

This study will be submitted for presentation at national or international surgical conferences. Manuscript(s) will be prepared following close of the study.

**Author affiliations**
¹Division of Cancer Sciences, School of Medical Sciences, Faculty of Biology, Medicine and Health, University of Manchester, Manchester, UK
²Manchester Cancer Research Centre, Manchester, UK
³Hepatobiliary Surgery, Central Manchester University Hospitals NHS Foundation Trust, Manchester, UK
⁴Oncology, Christie NHS Foundation Trust, Manchester, UK
⁵Institute of Cardiovascular Sciences, University of Manchester, Manchester, UK
⁶Centre for Imaging Sciences, University of Manchester, Manchester, UK
⁷Surgery, Christie NHS Foundation Trust, Manchester, UK
⁸Public Programmes Team, Research and Innovation Division, Manchester University NHS Foundation Trust, Manchester, UK
⁹Cancer Patient and Public Advisory Group, NIHR Manchester Biomedical Research Centre, Manchester, UK

**Contributors** The original study concept was conceived by JWV, DOR and AGR. The study design was developed by KLP, MB, AGR, DOR and CB. The imaging protocols were developed by SRW, JHW and WKL. Study set up was carried out by KLP, LM, AGR, DOR, KC, MB, JHW, SRW and WKL. The present protocol was written by DOR, AGR and KLP. All authors contributed to the revision and preparation of this manuscript.

**Funding** This research has received funding from the following sources: Humedics GMBH; National Institute for Health Research Manchester Biomedical Research Centre; North West National Health Service (NHS) Innovation service; The Christie NHS Foundation Trust charitable funds; The Royal College of Surgeons of England Fellowship scheme.

**Competing interests** None declared.

**Patient consent for publication** Obtained.

**Provenance and peer review** Not commissioned; externally peer reviewed.

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
