## [Reviewer comments · BMJ Open]

ARTICLE DETAILS

TITLE (PROVISIONAL)	A prospective study of Change in Liver Function and Fat in patients with colorectal liver metastases undergoing pre-operative chemotherapy: protocol for the CLiFF study
AUTHORS	Parmar, Kat; O'Reilly, Derek; Valle, Juan; Braun, Michael; Naish, Jo; Williams, Steve; Lloyd, William; Malcomson, Lee; Cresswell, Katharine; Bamford, Colin; Renehan, Andrew

VERSION 1 – REVIEW

REVIEWER	Wataru Gono Radiology, The University of Tokyo, Japan
REVIEW RETURNED	18-Dec-2018

GENERAL COMMENTS	This manuscript explains a study protocol that aims to evaluate the effects of pre-operative chemotherapy upon liver function and fat using LiMAX, magnetic resonance (MR) modified Dixon method, and MR spectroscopy. These methods are non-invasive and enable repeated assessments of liver function and fat. Abstract: OK Background: Redundant, but well-documented. Methods - Data Collection: Please explain imaging protocols of MR modified Dixon method and MR spectroscopy (TE, TR, flip angles, etc. Do you make any fat-images? Please indicate who will assess those images? (two or more board-certified radiologists?) How do you evaluate liver fat on MR images? – Qualitatively or quantitatively? Manually or automatically? On fat-image or on both in-phase and out-of-phase images? Thank you for letting me participate in this review
---

REVIEWER	Shishir K. Maithel MD FACS Division of Surgical Oncology Winship Cancer Institute Emory University Atlanta, GA USA
REVIEW RETURNED	13-Jan-2019

GENERAL COMMENTS	The authors describe a very important trial. Liver metastases from colorectal cancer represent the most common indication for liver resection. Patients who undergo resection of their liver metastases have repeatedly been demonstrated to have superior overall survival
---

	compared to those who are not candidates for resection. Often, a perioperative chemotherapy regimen and treatment strategy is employed. There are, however, data that suggest chemotherapy is not indicated for certain clinical situations. Furthermore, given the potential liver toxicity from chemotherapy and the potential clinical implications that the toxicity can have in terms of operative candidacy and postoperative complications and outcomes, it is imperative to fully understand these changes that are induced in the liver from the most common chemotherapeutic agents that we use. A better and accurate understanding of these changes will help to guide the duration of preoperative chemotherapy. The authors describe an elegant study that will assess histologic changes in the liver using NON-INVASIVE methods and will compare their findings to the actual findings on microscopic histologic examination after patients undergo resection. Information gathered from this study will be important to future trial design and clinical care.
--	--

VERSION 1 – AUTHOR RESPONSE

Reviewer(s) Reports:

Reviewer: 1

Reviewer Name: Wataru Gono

Institution and Country: Radiology, The University of Tokyo, Japan

Please state any competing interests or state 'None declared': None declared

Please leave your comments for the authors below

This manuscript explains a study protocol that aims to evaluate the effects of pre-operative chemotherapy upon liver function and fat using LiMAX, magnetic resonance (MR) modified Dixon method, and MR spectroscopy. These methods are non-invasive and enable repeated assessments of liver function and fat.

Abstract: OK

Background: Redundant, but well-documented.

Methods - Data Collection:

Please explain imaging protocols of MR modified Dixon method and MR spectroscopy (TE, TR, flip angles, etc. Do you make any fat-images?

Please indicate who will assess those images? (two or more board-certified radiologists?)

How do you evaluate liver fat on MR images? – Qualitatively or quantitatively? Manually or automatically? On fat-image or on both in-phase and out-of-phase images?

Authors reply and actions: On page 9 we have extensively revised and added the MR methodology as follows:

“MRS

Single volume 1H spectra are acquired using Stimulated Echo Acquisition Mode (STEAM) spectroscopy (TR = 2000 ms, TE = 10 ms, FA = 90°). 16 dynamic scans are acquired without averaging for a 15 mm³ voxel of interest (VOI) with a 32 s duration. This is repeated three times with VOIs positioned to avoid CLM and partial volume averaging with large blood vessels and bile ducts.

mDixon

A six-echo 3D gradient echo sequence is used, with first TE at 0.92 ms and a delta of 0.7 ms (TR = 5.3 ms, FA = 5°). Full liver coverage is acquired with 67 axial slices and 3 mm² in-plane and 6 mm through plane resolution, interpolated to 2 mm² in-plane and 3 mm through plane. Parallel acquisition used with a SENSE factor of 1.5 in the AP plane. The full volume is acquired in a single breath hold for a duration of 12 seconds. Quantitative fat fraction maps are calculated using a dedicated scanner software package. A representative fat fraction for each subject is calculated using the average from three liver-tissue regions of interest positioned on a central slice, using the same criteria as for the spectroscopy.”

And on page 13, we write:

“The study MR images will not be reported by a clinical radiologist.”

Thank you for letting me participate in this review

Reviewer: 2

Reviewer Name: Shishir K. Maithel MD FACS

Institution and Country: Division of Surgical Oncology

Winship Cancer Institute

Emory University

Atlanta, GA

USA

Please state any competing interests or state ‘None declared’: None declared